# Single Transmitter Direction Finding Using a Single Moving Omnidirectional Antenna

**DOI:** 10.3390/s22239208

**Published:** 2022-11-26

**Authors:** Guy Eliyahu, Amnon Menashe Maor, Roei Meshar, Reem Mukamal, Anthony J. Weiss

**Affiliations:** 1Elisra-Elbit Ltd., Holon 5885118, Israel; 2School of Electrical Engineering, Tel Aviv University, Tel Aviv 6997801, Israel

**Keywords:** direction finding (DF), Angle-of-Arrival (AOA), Direction of Arrival (DOA), Single Sensor, Cramér–Rao Lower Bound (CRLB)

## Abstract

Traditional direction-finding systems are based on processing the outputs of multiple spatially separated antennas. The impinging signal Angle-of-Arrival (AOA) is estimated using the relative phase and amplitude of the multiple outputs that are sampled simultaneously. Here, we explore the potential of a single moving antenna to provide useful direction finding of a single transmitter. If the transmitted signal frequency is steady enough during the collection of data, a single antenna can be moved while tracking the phase changes to provide an Angle-of-Arrival measurement. The advantages of a single-antenna sensor include the sensor size, the lack of a need for multiple-receiver synchronization in time and frequency, the lack of mutual antenna coupling, and the cost of the system. However, a single-antenna sensor requires an accurate knowledge of its position during the data collection and it is challenged by transmitter phase instability, signal modulation, and transmitter movement during the measurement integration time. We analyze the performance of the proposed sensor, support the analysis with simulations and finally, present measurements performed by hardware configured to check the validity of the proposed single-antenna sensor.

## 1. Introduction

This manuscript explores the possibility of estimating Angle-of-Arrival (azimuth and elevation) using only a single moving antenna. The method is analyzed theoretically via mathematical analysis and by comparison to the Cramér–Rao Lower Bound (CRLB) [1,2,3]. Simulations verify that the proposed algorithm’s performance coincides with the theoretical bound asymptotically, as the size of the error vector decreases. The idea is further explored and the concept is proved via the construction of a system using commercially available components.

Direction finding can be used for tracking and for localization of emitters [4]. Existing direction finding (DF) systems are based on an array of multiple spatially separated antennas [5,6]. The outputs of the antennas are sampled simultaneously and the multiple sample sequences are processed in order to estimate the Angle-of-Arrival of possibly multiple signals [7,8,9]. Since the antennas are spatially separated, in general the signal observed at each antenna has a different phase. This phase difference is exploited in order to estimate the AOA. Obviously, a single *stationary* antenna cannot obtain information for AOA estimation. However, a moving antenna might be envisioned as an array of antennas and a single receiver that samples the antennas sequentially. Multiple antennas with fewer receivers are discussed in s [10,11,12] and in many other publications. Recently, it was suggested in [13], to use a single-antenna AOA estimation for ultra-wideband transmissions of two synchronized antennas. In [14], the authors discuss a moving single antenna that exploits AOA of multipath for positioning.

Reference [15] introduces a supervised method for estimating both the location and velocity of a moving acoustic source using a single microphone based on a manifold learning approach. Reference [16] proposes a single antenna whose radiation pattern changes along time by using four rotating blades near the antenna. A single RF channel with an array of eight antennas is discussed in [17]. The single channel is connected to each of the antennas in the array, one at a time, by a switch. Reference [18] discusses a single antenna with several radiation patterns obtained by switching parasitic elements. Thus, each AOA is associated with a different set of antenna power outputs that enable direction finding.

None of the above works discuss a general AOA method that is based on a single moving antenna. The advantages of a single-antenna sensor include small sensor size, the lack of a need for multiple-receiver synchronization, in time and frequency, the lack of mutual-antenna coupling [19], and the cost of the system. However, a single-antenna sensor requires an accurate knowledge of its position during the data collection and it is challenged by transmitter-phase instability, signal modulation, and transmitter movement during the measurement integration time. We analyzed the performance of the proposed sensor, supported the analysis with simulations, and finally, presented measurements performed by hardware configured to check the validity of the proposed single-antenna sensor.

Section 2 presents the mathematical model, derives the problem to be solved, and proposes a suitable algorithm. Section 3 derives the CRLB under various sources of errors including additive noise, phase noise, and velocity errors. Section 4 describes the results of computer simulations using the proposed algorithm. A comparison of the simulation errors and the CRLB shows that the algorithm is statistically efficient. Section 5 describes the hardware and the field trials that demonstrate that the concept is valid. Finally, Section 6 presents the conclusions.

We use boldface lower case font (e.g., x) to denote column vectors, boldface upper case fonts (e.g., X) to denote matrices, ≜ denotes ‘equal by definition’,$\left\|\;\circ \;\right\|$ denotes the $\ell^{2}$ norm, ${\left(\circ \right)}^{T}$ denotes transposition, and ⊗ is the Kronecker product [20].

## 2. Mathematical Model

Consider a transmitter whose unknown position is given by three Cartesian coordinates denoted by the 3×1 time-dependent vector,
(1)p(t)=p0+wt+0.5at2∈ℝ3×1;       0≤t≤Tm
where t is the time variable, p0∈ℝ3×1 is the transmitter unknown position at time t=0, and w,a∈ℝ3×1 are the transmitter unknown velocity and unknown acceleration vectors, respectively. Finally, Tm is the measurement integration time.

Further, consider a moving antenna whose position along time is described by three Cartesian coordinates denoted by the 3×1 time-dependent vector q(t)∈ℝ3×1. Let the unit vector pointing from the antenna to the transmitter be denoted by u∈ℝ3×1. This vector is also known as the Line of Sight (LOS) vector. Since, in general, the distance between the antenna and the transmitter is many orders of magnitude larger than the distances traveled by the antenna and the transmitter during a single measurement, the vector u is, for all practical purposes, constant along time. Let the transmitted signal be given by
(2)sT(t)=A(t)ej[ωct+ϕ]∈ℝ
where A(t) is some unknown amplitude modulation, ωc is the carrier frequency in [radians/second], and ϕ is some unknown constant phase. If the signal is also phase modulated, we assume that the phase change due to the modulation is negligible during the measurement. The propagation from the transmitter to the moving antenna introduces an attenuation given by α, and a delay τ(t) that is proportional to the distance between the two. Thus, the received signal is given by
(3)sR(t)=αA(t−τ(t))ej[ωc(t−τ(t))+ϕ].

We assume that the signal is sampled uniformly at time instances $t_{i}=iT_{s};\;i=\left\{0,1\cdots ,N\right\}$.

Note that the received signal phase is given by
(4)$$\psi (t_{i})≜\omega_{c}(t_{i}-\tau (t_{i}))+\phi .$$

Since ϕ is unknown, it can be eliminated by taking the difference of every two consecutive phase samples, which yields
(5)$$\frac{1}{T_{s}}\left[\psi (t_{i+1})-\psi (t_{i})\right]=\omega_{c}-\omega_{c}\frac{\left[\tau (t_{i+1})-\tau (t_{i})\right]}{T_{s}}.$$

This motivates us to find the dependence of
(6)$$\begin{array}{l} \Delta \tau ({\bar{t}}_{i,i+1})≜\frac{1}{T_{s}}\left[\tau (t_{i+1})-\tau (t_{i})\right]\\ \;\;\;\;\;\;{\bar{t}}_{i+1,i}≜\frac{t_{i+1}+t_{i}}{2} \end{array}$$
on the unit vector u that points from the antenna to the transmitter. Towards this end, note that
(7)$$\tau (t_{i})=\frac{1}{c}\left\|p(t_{i})-q(t_{i})\right\|$$
where c is the signal propagation speed. We approximate (6) by the derivative of τ(t):(8)$$\begin{array}{l} \dot{\tau }(t)=\frac{1}{c}\frac{\partial }{\partial t}\sqrt{\sum_{j=1}^{3}{\left(p_{j}(t)-q_{j}(t)\right)}^{2}}\\ \;\;\;\;\;\;=-\frac{1}{c}\sum_{j=1}^{3}\frac{p_{j}(t)-q_{j}(t)}{\left\|p(t)-q(t)\right\|}\left[{\dot{q}}_{j}(t)-{\dot{p}}_{j}(t)\right]\\ \;\;\;\;\;\;=-\frac{1}{c}{\left[\dot{q}(t)-w-at\right]}^{T}u. \end{array}$$

Note that in (8) we exploited the fact that the elements of u are given by
(9)$$u_{j}=\frac{p_{j}(t)-q_{j}(t)}{\left\|p(t)-q(t)\right\|}\;\;\;\;\;j\in \{1,2,3\}$$

Since $\dot{q}(t)$ is the velocity vector of the receiver, we define $v(t)≜\dot{q}(t)$. Substituting (8) in (5) we obtain the compact expression
(10)$$\begin{array}{l} \gamma ({\bar{t}}_{i+1,i})≜\frac{1}{T_{s}}\left[\psi (t_{i+1})-\psi (t_{i})\right]-\omega_{c}\\ \;\;\;\;\;\;\;\;\;\;\;≅k{\left[v({\bar{t}}_{i+1,i})-w-a{\bar{t}}_{i+1,i}\right]}^{T}u;\\ \;\;\;\;\;\;\;\;\;k≜\frac{\omega_{c}}{c}=\frac{2\pi }{\lambda };\;\;\;\;\;i=0,1,\cdots ,N-1 \end{array}$$
where k is the wave number and λ is the signal wavelength. We assume that the signal frequency $\omega_{c}$ is known and, therefore, we can use $\gamma ({\bar{t}}_{i+1,i})$ in (10) to find u. In real life, the ideal (10) should be modified to include various errors as discussed in the sequel. Adding an error term, $e(t_{i})$, to (10) we can write
(11)$$\gamma ({\bar{t}}_{i+1,i})=k{\left[v({\bar{t}}_{i+1,i})-w-a{\bar{t}}_{i+1,i}\right]}^{T}u+e({\bar{t}}_{i+1,i})$$

Although we want to estimate u, which has three elements, only two elements are independent. To see this note that
(12)$$u={\left[\cos\theta_{1}\cos\theta_{2},\;\sin\theta_{1}\cos\theta_{2},\;\sin\theta_{2}\right]}^{T}$$
where $\theta_{1}$ is the azimuth angle, measured anticlockwise from the positive x-axis in the x–y plane, and $\theta_{2}$ is the elevation angle, measured anticlockwise from the x-y plane towards the z-axis.

Equation (11) can be formulated by a vector-matrix equation. Define
(13)$$\begin{array}{l} \;f≜{\left[\gamma ({\bar{t}}_{1,0}),\cdots ,\gamma ({\bar{t}}_{N,N-1})\right]}^{T}\\ \;e≜{\left[e({\bar{t}}_{1,0}),\cdots ,e({\bar{t}}_{N,N-1})\right]}^{T}\\ V≜k{\left[\begin{array}{ccc} v({\bar{t}}_{1,0}) & ,\cdots , & v({\bar{t}}_{N,N-1}) \end{array}\right]}^{T}\\ \;a≜kw^{T}u\\ \;b≜ka^{T}u. \end{array}$$

Now (11) can be written in a compact form,
(14)$$\begin{array}{l} f=Vu-a1-bt+e\\ 1≜{\left[1,1,\cdots ,1\right]}^{T}\\ t≜{\left[{\bar{t}}_{1,0},{\bar{t}}_{2,1},\cdots ,{\bar{t}}_{N,N-1}\right]}^{T}. \end{array}$$

The unknowns in (14) are u,a,b and, therefore, it is more convenient to write it as
(15)$$\begin{array}{l} \;\;f=Wx+e\\ W≜\left[\begin{array}{ccc} V & -1 & -t \end{array}\right]\\ \;\;x≜{\left[\begin{array}{ccc} u^{T} & a & b \end{array}\right]}^{T}. \end{array}$$

Since u is a unit vector, meaning that $u^{T}u=1$, we introduce the matrix:(16)$$G≜\left[\begin{array}{ccc} I_{3} & 0 & 0\\ 0 & 0 & 0\\ 0 & 0 & 0 \end{array}\right]$$
where $I_{3}$ is the 3×3 identity matrix and the constraint
(17)$$x^{T}Gx=1.$$

We assume that the expectation of the error vector e is zero and its covariance is given by
(18)$$E\left\{ee^{T}\right\}=\Sigma .$$

Then, the estimator can be defined as a Weighted Least Squares Estimator (WLSE) with constraints:(19)$$\begin{array}{l} \mathrm{minimize}\;\;{\left[f-Wx\right]}^{T}\Sigma^{-1}\left[f-Wx\right]\\ \mathrm{subject}\;\mathrm{to}:\;x^{T}Gx=1. \end{array}$$

Obviously, this is a quadratic minimization problem with equality quadratic constraints that can be solved by an interior-point algorithm [21]. Note that if the distribution of the error vector e is Gaussian, the solution of (19) is the maximum likelihood estimator [22], whose error covariance is expected to approach the Cramér–Rao Lower Bound, under asymptotic conditions (high SNR or long integration time).

In summary, we want to estimate $\theta_{1},\theta_{2}$ (both embedded in u) given the vector f and the matrices W,G,Σ.

## 3. Derivation of the CRLB

Under the assumption that the error vector, e, is Gaussian, we can derive the Cramér–Rao Lower Bound by using closed-form expressions from [22]. The 4×4 Fisher Information Matrix (FIM) is given by
(20)$$\begin{array}{l} J_{i,j}=g_{i}^{T}W^{T}\Sigma^{-1}Wg_{j};\;\;\;\;\;\;\;\;i,j\in \left\{1,2,3,4\right\}\\ g_{1}^{T}≜[-s_{1}c_{2},c_{1}c_{2},0,0,0];\\ g_{2}^{T}≜[-c_{1}s_{2},-s_{1}s_{2},c_{2},0,0]\\ g_{3}^{T}≜[0,0,0,1,0]\\ g_{4}^{T}≜[0,0,0,0,1]\\ c_{k}≜\cos\theta_{\ell };\;\;\;\;\;\;\ell =1,2\\ s_{k}≜\sin\theta_{\ell };\;\;\;\;\;\;\;\ell =1,2 \end{array}$$
where the indices i,j refer to the entries of the vector of parameters to be estimated: $\left[\theta_{1},\theta_{2},a,b\right]$. The CRLB is given by the inverse of the FIM. Thus,
(21)$$\begin{array}{c} CRLB={\left[E^{T}W^{T}\Sigma^{-1}WE\right]}^{-1}\\ E≜[g_{1},g_{2},g_{3},g_{4}] \end{array}$$
which is a closed-form expression for the CRLB. Note in passing that the CRLB involves the inversion of a matrix. The matrix to be inverted is not guaranteed to be non-singular in all cases. In particular, if the sensor moves on a straight line the matrix becomes singular which means that the azimuth and elevation angles cannot be estimated simultaneously.

We now turn to find expressions for the error covariance, Σ, under different circumstances that include additive noise, sensor velocity errors, and phase noise.

### 3.1. Additive Noise

We now examine the effect of additive noise on the error vector introduced in (11). Modifying (3) to include additive noise we obtain
(22)$$s_{Rn}(t_{i})=s_{R}(t_{i})+n(t_{i}).$$

The noisy phase of the signal is now given by
(23)$$\psi_{n}(t_{i})=\arctan\frac{{\tilde{s}}_{R}(t_{i})+\tilde{n}(t_{i})}{{\bar{s}}_{R}(t_{i})+\bar{n}(t_{i})}$$
where the bar $\bar{\circ }$ represents the real part and the tilde $\tilde{\circ }$ represents the imaginary part. The error in the signal phase due to small noise is given by
(24)Δψ(ti)≜ψn(ti)−ψ(ti)≈ −s˜R(ti)+n˜(ti)n¯(ti)s¯R(ti)+n¯(ti)2+s˜R(ti)+n˜(ti)2+s¯R(ti)+n¯(ti)n˜(ti)s¯R(ti)+n¯(ti)2+s˜R(ti)+n˜(ti)2    ≅−1sR(ti)2ImsR*(ti)n(ti).

The result in (24) is a random variable with the first two moments given by
(25)$$\begin{array}{l} E\left\{\Delta \psi (t_{i})\right\}=0\\ E\left\{\Delta \psi^{2}(t_{i})\right\}=\frac{\sigma^{2}}{2{\left|s_{R}(t_{i})\right|}^{2}}=\frac{1}{2SNR} \end{array}$$
where SNR denotes signal-to-noise ratio. This result is valid for zero-mean, circular, symmetric, complex noise [23] with a variance of $\sigma^{2}$. Exploiting (10), we can write
(26)$$\begin{array}{l} \delta \gamma ({\bar{t}}_{i+1,i})≜\gamma_{n}({\bar{t}}_{i+1,i})-\gamma ({\bar{t}}_{i+1,i})\\ \;\;\;\;\;\;\;\;\;\;\;\;\;=\frac{\psi_{n}(t_{i+1})-\psi_{n}(t_{i})-\psi (t_{i+1})+\psi (t_{i})}{T_{s}}\\ \;\;\;\;\;\;\;\;\;\;\;\;\;=\frac{\Delta \psi (t_{i+1})}{T_{s}}-\frac{\Delta \psi (t_{i})}{T_{s}} \end{array}$$
which leads to
(27)$$\begin{array}{c} E\left\{e_{i}\right\}=E\left\{\delta \gamma ({\bar{t}}_{i+1,i})\right\}=0\\ \;\;\;\;\;\;\;E\left\{e_{i}^{2}\right\}=E\left\{\delta \gamma^{2}({\bar{t}}_{i+1,i})\right\}=\frac{1}{T_{s}^{2}SNR}\\ \;\;\;\;\;E\left\{e_{k}e_{k+1}\right\}=E\left\{\delta \gamma ({\bar{t}}_{k+1,k})\delta \gamma ({\bar{t}}_{k+2,k+1})\right\}\\ \;\;\;\;\;\;\;\;\;\;\;\;\;\;\;=\frac{1}{T_{s}^{2}}E\left\{\left[\Delta \psi (t_{k+1})-\Delta \psi (t_{k})\right]\left[\Delta \psi (t_{k+2})-\Delta \psi (t_{k+1})\right]\right\}\\ \;\;\;\;\;\;\;\;=\frac{1}{T_{s}^{2}}E\left\{-\Delta \psi^{2}(t_{k+1})\right\}=-\frac{1}{2T_{s}^{2}SNR}. \end{array}$$

Thus, the covariance of the error vector associated with additive, zero-mean, circular, symmetric, complex noise is given by the 3-diagonal Toeplitz matrix:(28)$$\begin{array}{c} \Sigma =\frac{1}{T_{s}^{2}SNR}D\\ D≜\left[\begin{array}{cccc} 1 & -0.5 & 0 & 0\\ -0.5 & 1 & \ddots & 0\\ 0 & \ddots & \ddots & -0.5\\ 0 & 0 & -0.5 & 1 \end{array}\right]. \end{array}$$

Thus, the CRLB in the presence of Additive White Gaussian Noise (AWGN) is given by
(29)$$\begin{array}{l} CRLB_{n}=\frac{1}{T_{s}^{2}\cdot SNR\cdot k^{2}}{\left[E^{T}{\tilde{W}}^{T}D^{-1}\tilde{W}E\right]}^{-1}\\ \;\;\;\;=\frac{\lambda^{2}}{T_{s}^{2}\cdot SNR\cdot {\left(2\pi \right)}^{2}}{\left[E^{T}{\tilde{W}}^{T}D^{-1}\tilde{W}E\right]}^{-1}\\ \;\;\tilde{W}≜\frac{1}{k}W. \end{array}$$

Note that $\frac{1}{k}VT_{s}/\lambda$ can be interpreted as the distance traveled within one sampling period in terms of wavelengths (normalized sensor speed). As expected, the error covariance is inversely proportional to the signal-to-noise ratio and to the squared sensor speed.

### 3.2. Sensor Velocity Errors

If the sensor path during the DF measurement is obtained with errors, it affects the sensor velocity estimates. We obtain from (14) that the error vector is given by
(30)$$\begin{array}{l} \Delta V=V_{m}-V\\ e_{v}=\Delta Vu \end{array}$$
where $V_{m}$ is the measured (estimated) velocity matrix and V is the true velocity. We assume that the entries of ΔV are independent, identically distributed (i.i.d.), zero-mean, and with standard deviation of $\sigma_{v}$. Thus,
(31)$$\begin{array}{c} E\left\{e_{v}\right\}=0\\ \;\;\;\;\;E\left\{e_{v}e_{v}^{T}\right\}=E\left\{\Delta Vuu^{T}\Delta V^{T}\right\}\\ \;\;\;\;\;\;\;\;\;\;\;\;\;\;\;\;=(u^{T}⊗I_{N})E\left\{\Delta V(:)\Delta V^{T}(:)\right\}(u⊗I_{N})\\ \;\;\;\;\;\;\;\;\;\;\;\;\;\;\;\;=\sigma_{v}^{2}(u^{T}⊗I_{N})(u⊗I_{3})=\sigma_{v}^{2}(u^{T}u⊗I_{N})\\ \;\;\;\;\;\;\;\;\;=\sigma_{v}^{2}(1⊗I_{N})=\sigma_{v}^{2}I_{N} \end{array}$$
where ΔV(:) is the 3N×1 vector formed by concatenating the columns of ΔV.

Thus, the CRLB for sensor velocity errors is given by
(32)$$CRLB_{V}=\sigma_{v}^{2}\frac{\lambda^{2}}{{\left(2\pi \right)}^{2}}{\left[E^{T}{\tilde{W}}^{T}\tilde{W}E\right]}^{-1}.$$

Observe that, according to the bound, the AOA error variance is proportional to the velocity error variance, as expected.

### 3.3. Phase Noise

Assume now that the phase observed at the sensor is affected by phase noise that is generated by the circuits of the transmitter and of the receiver.

Now, $\psi (t_{i})$ in (4) becomes
(33)$$\hat{\psi }(t_{i})=\psi (t_{i})+\delta \psi (t_{i})$$
where $\delta \psi (t_{i})$ denotes the phase error at time $t_{i}$. Following the derivation of (14), we learn that the error vector, $e_{p}$ is now
(34)$$e_{p}=\frac{1}{T_{s}}{\left[(\delta \psi (t_{1})-\delta \psi (t_{0})),\cdots ,(\delta \psi (t_{N})-\delta \psi (t_{N-1}))\right]}^{T}.$$

Assuming that the samples of the phase error are i.i.d., zero-mean, and with variance of $\sigma_{p}^{2}$, we obtain that
(35)$$\begin{array}{c} E\left\{e_{p}\right\}=0\\ \;E\left\{e_{p}e_{p}^{T}\right\}=\frac{2}{T_{s}^{2}}\sigma_{p}^{2}D. \end{array}$$

Thus, the CRLB in the presence of phase noise becomes,
(36)$$CRLB_{p}=\frac{2\sigma_{p}^{2}\lambda^{2}}{T_{s}^{2}{\left(2\pi \right)}^{2}}{\left[E^{T}{\tilde{W}}^{T}D^{-1}\tilde{W}E\right]}^{-1}.$$

Observe that the bound is proportional to the variance of the phase error, as expected.

## 4. Simulation Results

To verify the theoretical results, we performed Monte-Carlo computer simulations. The antenna motion during a single DF measurement is depicted in Figure 1.

The antenna moves on the surface of a sphere whose radius is 0.5 m. There are 200 sampling points during a single DF measurement. The measurement is completed within 0.6252 s. The sampling rate is 318.3 samples per second. The center frequency of the transmitter is 1 GHz, its location is at coordinates [x,y,z]=[1000,200,200] meters, its velocity is $\left[v_{x},v_{y},v_{z}]=[20,0,0\right]$ [meter/second], and its acceleration is $\left[a_{x},a_{y},a_{z}]=[2,0,0\right]$ [m/s^2^]. For each simulated point we used 100 Monte-Carlo experiments.

Figure 2 shows the AOA standard deviation versus the SNR. The red color relates to azimuth estimation and the blue color relates to elevation estimation. The solid lines show the CRLB while the dashed lines with circles show the simulation results. It can be seen that for SNR above 10 dB the simulations match the lower bound. For low SNR the errors are much higher than the bound.

Figure 3 shows the AOA standard deviation versus the standard deviation of the velocity error. The error units are in percent of the average velocity. We observe a rather good fit between the bound and the simulation results.

Figure 4 shows the AOA standard deviation versus the phase error standard deviation. Again, we see a rather good match between the simulations and the bound.

In all figures we see that the azimuth error is smaller than the elevation error. This is due to the antenna path that has many more sampling points that contribute to azimuth accuracy than points that contribute to elevation accuracy. Changing the path causes changes in the accuracy.

## 5. Hardware Proof of Concept

In order to verify experimentally the proposed DF method, we implemented a hardware proof of concept led by a team from Elisra-Elbit Ltd. The transmitter is a signal generator (Agilent 8648D) connected to an omnidirectional wideband antenna (Cisco). Figure 5 shows the signal generator and transmitting antenna. The receiving antenna was an omnidirectional antenna connected to a tiny Software-Defined Radio (SDR) manufactured by Nooelec. The receiving antenna and an Inertial Measurement Unit (IMU) were mounted on a tip of a rod, see Figure 6. The IMU is manufactured by Inertial-Sense (uINS-3 ragged.) The base of the rod was fixed to the floor and its other tip moved on a sphere of radius 1 m (the length of the rod) in a random path. See Figure 7 for a single realization of the antenna path. The color indicates the time passed from the start of the movement. Note that the path is confined to a sphere, as expected. Figure 8 shows the AOA standard deviation (STD) and the bias versus the actual azimuth.

The samples from the SDR and the IMU were collected and processed. One of the challenges is to time synchronize the IMU output stream with the SDR output stream. This challenge can be mitigated by time adjustment that minimizes the AOA variability along time. Since the experiments were not performed in a sterile environment (an anechoic chamber, for example), it is reasonable to expect multipath. We did not have the means to verify the amount of multipath interference. Note, however, that Figure 8 shows that the standard deviation and the bias of the measured AOA vary strongly with the actual positions of the receiver and transmitter. This suggests that multipath and related effects were a major source of error.

Finally, we examined the effect of integration time, $T_{m}$, on the AOA standard deviation. See Figure 9. The longer the integration time, the lower the AOA standard deviation.

## 6. Conclusions

In this work, we demonstrated that the Angle-of-Arrival can be measured by a single moving antenna. For a single moving antenna, there are several challenges that include transmitter frequency stability, accurate antenna positioning knowledge during data collection, and transmitter unknown movement. We are aware of the fact that all these challenges cannot be perfectly mitigated. However, reasonable AOA measurements are still possible.

## Figures and Tables

**Figure 1 sensors-22-09208-f001:**
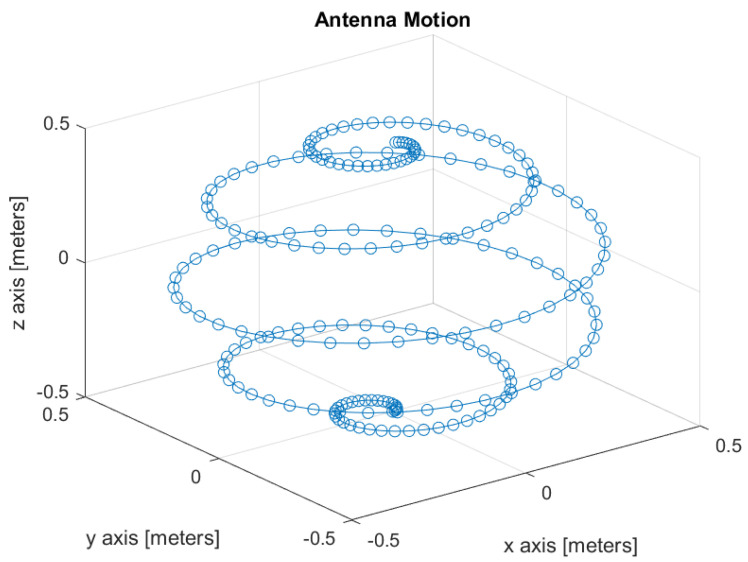
Antenna motion during a single measurement.

**Figure 2 sensors-22-09208-f002:**
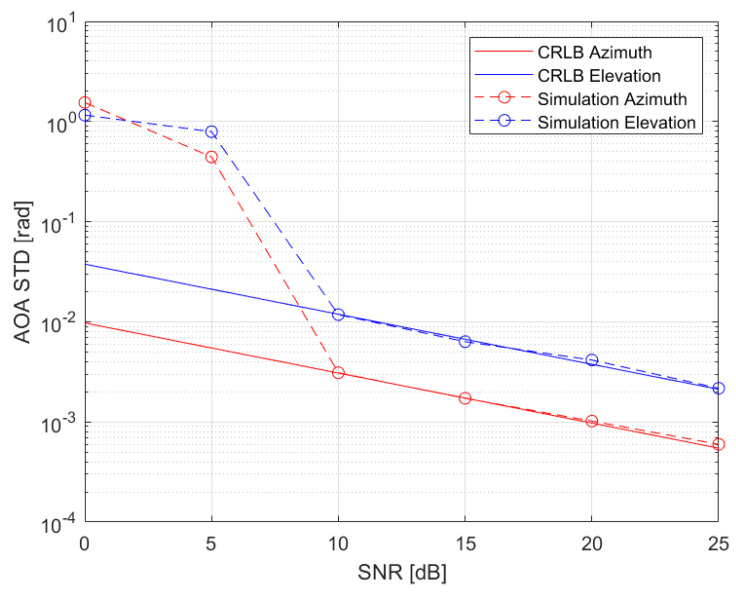
AOA standard deviation vs. SNR. Elevation is shown in blue, azimuth in red, CRLB in solid lines, and simulations in dashed lines with the circle marker.

**Figure 3 sensors-22-09208-f003:**
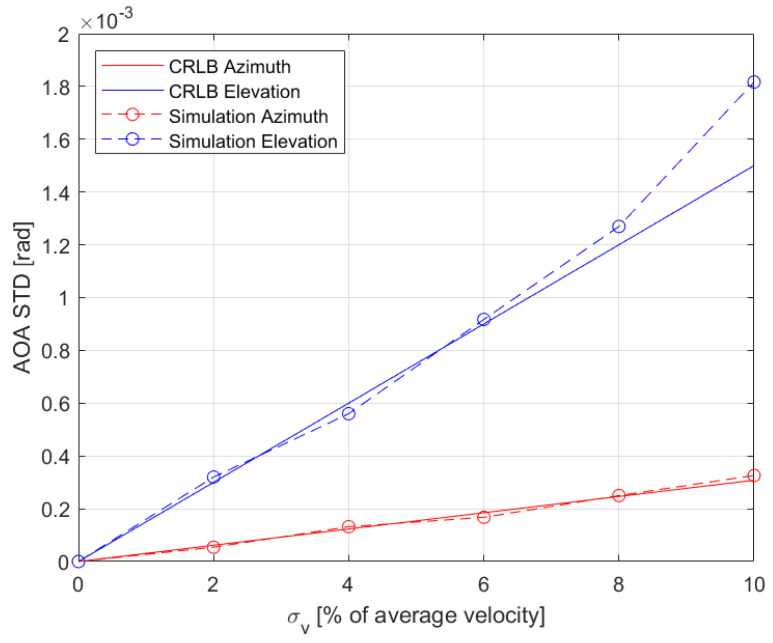
AOA standard deviation vs. velocity errors standard deviation. Elevation is shown in blue, azimuth in red, CRLB in solid lines, and simulations in dashed lines with the circle marker.

**Figure 4 sensors-22-09208-f004:**
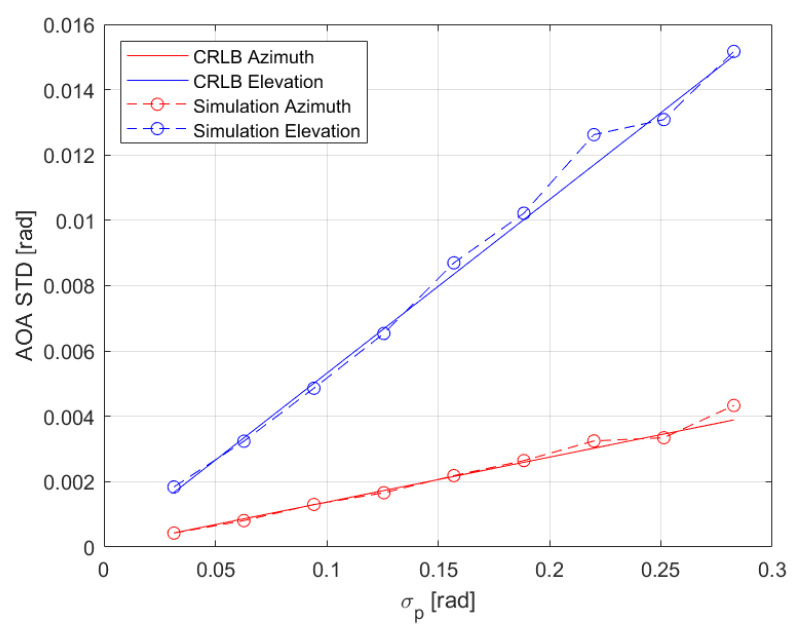
AOA standard deviation vs. phase errors standard deviation. Elevation is shown in blue, azimuth in red, CRLB in solid lines, and simulations in dashed lines with the circle marker.

**Figure 5 sensors-22-09208-f005:**
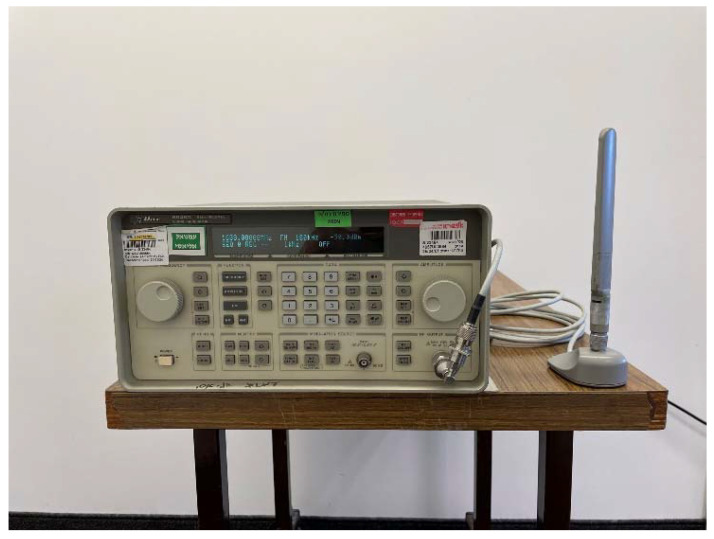
Signal generator and transmitting antenna.

**Figure 6 sensors-22-09208-f006:**
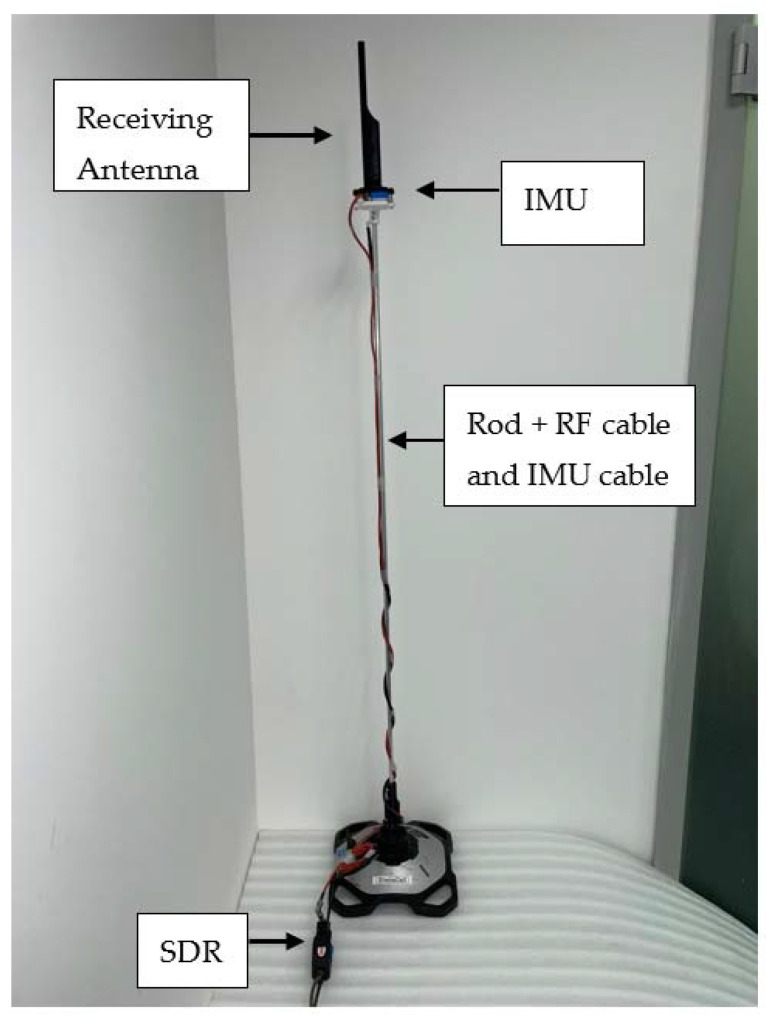
Receiving configuration including moving rod, receiving antenna, IMU, and SDR.

**Figure 7 sensors-22-09208-f007:**
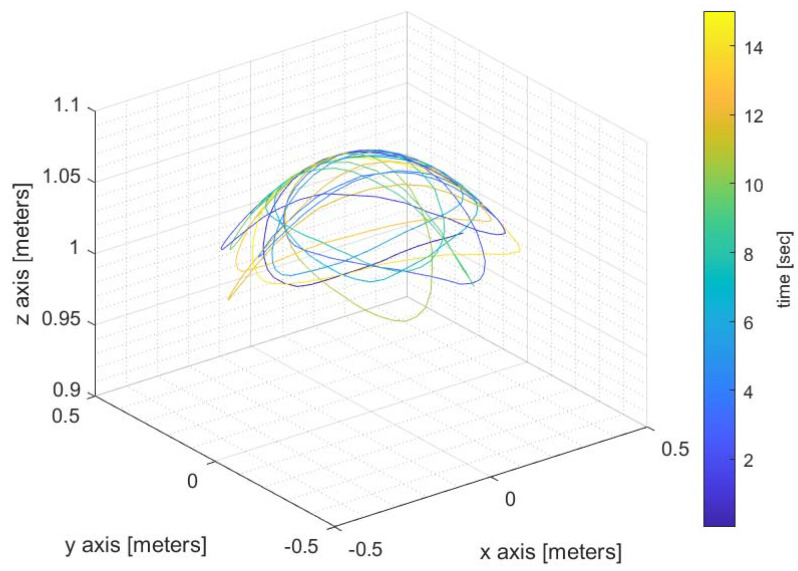
A single realization of the antenna random path. The color indicates time from start of movement.

**Figure 8 sensors-22-09208-f008:**
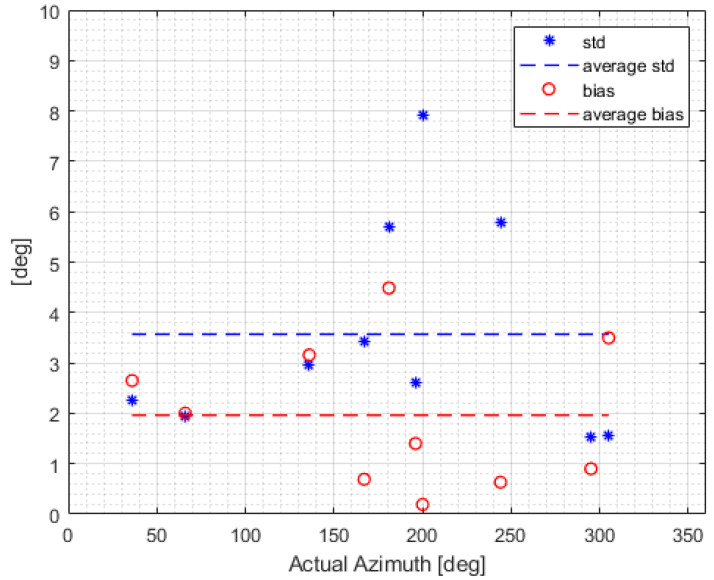
AOA standard deviation and bias versus the actual azimuth.

**Figure 9 sensors-22-09208-f009:**
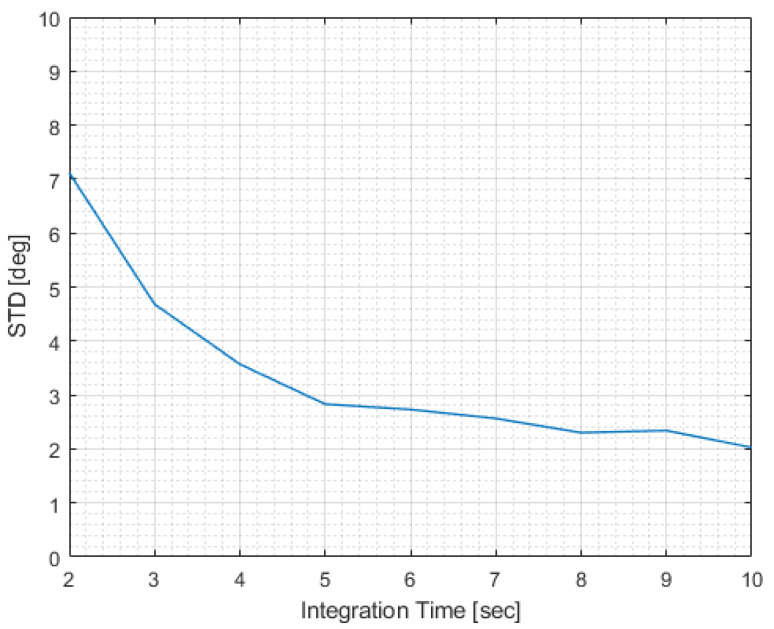
AOA standard deviation versus measurement integration time.

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
