# Peer review of "Single Transmitter Direction Finding Using a Single Moving Omnidirectional Antenna"

_sensors, 2022, doi:10.3390/s22239208_

Round 1

Reviewer 1 Report

First of all, a note: the manuscript has a formatting problem!
The text "Error! Reference source not found." is everywehere, instead of the opportune reference. This makes reading difficult.

Apart from that, the manuscript is good: the idea is solid, and the implementation seems reasonable. 

I really appreciate the montecarlo simulation and even more the experimental validation, which often is missing in thsese kind of work.

The results are reasonable, since the premise and the novelty of the paper.

Nevertheless, I suggest to study a little more the impact of the measurement noise in a real scenario, considering the impact of multipath, scattering and all the deterministic bu untreaceble mechanism impacting on the AoA accuracy.

As a side note, I suggest you add a few more references, especially for the reader who is new to this type of application.

Overall, the article is of sufficient quality for publication, in my opinion.

Reviewer 2 Report

  Authors have contributed excellent    work   entitled “Single Transmitter Direction Finding Using a Single Moving 2 Antenna” in terms of   DOA estimation   using   single  rotating antenna  or receiver and  provided   experimental  validation  as  proof of concept.   However ,  authors  have  to  justify   how  this  work  is  different   than  K. Ren, 2022.    Here presented proof of concept is   done in controlled environment   with stable point radiator to be    located.

K. Ren, “Direction finding using a single antenna with blade modulation,” IEEE Antennas Wireless Propag. Lett., early access, 2022.

1.       It  is suggested  to   present  the validation over  multiple points  over range  of 0 to  1800 

2.       Effect of   antenna polarization   has not been considered.

3.       A detailed  analysis  of  sources  of  errors in DOA  estimation   with   variation   of  elevation  angle  also  can be presented.

Reviewer 3 Report

The paper “Single Transmitter Direction Finding Using a Single Moving Antenna” is proposed a single antenna sensor. The impinging signal Angle-of-Arrival (AOA) is estimated using the relative phase and amplitude of the multiple outputs. By exploring the potential of a single moving antenna to provide useful Direction-Finding of a single transmitter. The advantages of a single antenna sensor include the sensor size, the lack of a need for multiple receiver synchronization, time and frequency, and the cost of a system. However, the following comments must be acknowledged.

·       The references are not as per the journal template, how can you the citation from [9] onwards? Why not from [1]?

·       The references should be sequenced so that the flow/decision of a paper will be accessed.

·       The introduction section is too short. Also, add the outline of a paper.

·       Some abbreviations are not reported properly such as IMU, etc.

·       Figures 2, 3, 4, 7, 8, and 9, need to be more clear.

Round 2

Reviewer 3 Report

Thank you for reporting the suggested comments. Although there are many typos and mistakes in equations, the missing quantities are, vector quantities, equality/inequality signs or belongs to, etc.

Figures 8 and 9, should be updated for y-axis values(up to 10 units).
